# The Disengagement of Visual Attention: An Eye-Tracking Study of Cognitive Impairment, Ethnicity and Age

**DOI:** 10.3390/brainsci10070461

**Published:** 2020-07-18

**Authors:** Megan Polden, Thomas D. W. Wilcockson, Trevor J. Crawford

**Affiliations:** 1Psychology Department, Lancaster University, Bailrigg, Lancaster LA1 4YF, UK; t.crawford@lancaster.ac.uk; 2School of Sport, Exercise and Health Sciences, Loughborough University, Epinal Way, Loughborough LE11 3TU, UK; t.wilcockson@lboro.ac.uk

**Keywords:** cognitive impairment, disengagement, attention, inhibition, “gap effect”, overlap, saccade

## Abstract

Various studies have shown that Alzheimer’s disease (AD) is associated with an impairment of inhibitory control, although we do not have a comprehensive understanding of the associated cognitive processes. The ability to engage and disengage attention is a crucial cognitive operation of inhibitory control and can be readily investigated using the “gap effect” in a saccadic eye movement paradigm. In previous work, various demographic factors were confounded; therefore, here, we examine separately the effects of cognitive impairment in Alzheimer’s disease, ethnicity/culture and age. This study included young (*N* = 44) and old (*N* = 96) European participants, AD (*N* = 32), mildly cognitively impaired participants (MCI: *N* = 47) and South Asian older adults (*N* = 94). A clear reduction in the mean reaction times was detected in all the participant groups in the gap condition compared to the overlap condition, confirming the effect. Importantly, this effect was also preserved in participants with MCI and AD. A strong effect of age was also evident, revealing a slowing in the disengagement of attention during the natural process of ageing.

## 1. Introduction

Alzheimer’s disease (AD) is a neurodegenerative disease that leads to profound cognitive impairment that includes changes in working memory [1,2]. AD is often diagnosed relatively late in the neuropathology of the disease, due to the lengthy and subjective assessments for the clinical diagnosis that are currently used. Subtle early impairments in executive function, attentional disengagement and other cognitive processes have been reported in people with AD [3,4]. Various attempts have been made to develop specific measures of attentional control in patients with AD [5,6,7,8]. However, these have included multiple cognitive operations or have not been grounded in neurophysiological research that has provided insights into the attentional disengagement. An exception is the work by Parasuraman and colleagues [9,10] using the Posner task. Posner [11] stated that orienting of attention comprised three distinct stages: (1) disengagement from the current stimulus; (2) movement to the new location; and (3) re-engagement with the target at the new location. According to the Posner model, attention must be disengaged from the current visual target, in order to facilitate an attentional shift from the old to the new target; just as in driving a car where you disengage from one gear, before moving the gear stick to a new gear. These distinct operations require multiple brain processes, with each contributing to the cost in terms of the overall processing time [12]. Parasuraman and colleagues [9,10] reported that the reaction times to a “valid” cue (that summoned automatic attention towards the target) was equivalent in the AD and control participants. In contrast, the reaction times to an “invalid” cue (that required disengagement of attention away from the cue) was substantially increased in the AD group. This suggested that the automatic orientation of attention was preserved in AD, but the ability to disengage attention was impaired. However, these results failed to be replicated in several laboratories [13,14].

Mounting research has demonstrated that the attentional operations used in eye tracking tasks can provide an early marker of neurodegenerative disease [15,16,17,18,19,20]. Importantly, eye movement abnormalities occur earlier than the more noticeable changes in memory, which present relatively late in the progression of the disease [21]. A dual saccadic paradigm is often used to evaluate attentional disengagement [22,23,24,25]. In the so-called “gap” condition, the fixation point is removed 200 ms prior to the presentation of the display target, resulting in a temporal “gap” between the offset of the fixation point and the presentation of the new target. This condition yields relatively fast reaction times due to the facilitation of the disengagement operation by the prior removal of the fixation point. In contrast, in the “overlap” condition, the fixation point remains for a period of time while the new target is displayed (see Figure 1a,b). Therefore, in this condition, there is a temporal overlap between the offset of the central fixation point and the onset of the target. The “gap effect” is measured by the difference in the mean saccadic reaction times between the gap and overlap conditions and yields an operational index of attentional disengagement [26,27]. A saccadic eye movement is triggered relatively early in comparison to situations where the fixation point remains visible with the peripheral target, as in the step or overlap conditions [18,25,28,29]. 

There has been relatively little research on the “gap effect” in patients with neurodegenerative disease. Prosaccades have the potential to assess attentional fluctuation in patients with neurodegenerative disease and offer an alternative to more traditional paper-based tests. The few studies that have been reported have yielded conflicting findings. For example, Yang et al. [30] reported in a sample of Chinese AD and mildly cognitively impaired (MCI) participants a substantially larger “gap effect” in comparison to healthy age-matched controls. In contrast, a recent study with Iranian participants revealed no difference in the prosaccade gap effect between AD participants and healthy controls [31]. Crawford et al. [17] found using a longitudinal design with European U.K. participants that the “gap effect” in AD was similar to that of the controls after a 12 month period. These differences could be due to a combination of methodological factors, including the participant populations, since to our knowledge, no study has contrasted different ethnicity groups within a single study design. It is important to examine the effect in various populations to determine the cultural validity of the gap effect. Restricting study populations to Western, educated, industrialized, rich and democratic (WEIRD) samples has contributed to the replicability crisis [32]. Eye movement characteristics have previously differed across ethnicity/cultural groups [33,34], therefore, comparisons across cultures is important.

In summary, this work is an exploration of attentional disengagement, to determine the potential mediating effects of: (a) cognitive impairment (contrasting European participants with AD, MCI and European healthy older participants); (b) healthy ageing (contrasting healthy young and older European participants); and (c) ethnicity/culture (contrasting older European older participants and older South Asian participants).

## 2. Materials and Methods

### 2.1. Participants 

The study included 32 participants with dementia caused by Alzheimer’s disease (AD: mean age = 74.32, SD = 7.57, age range = 59–86 years), 47 participants with mild cognitive impairment (MCI: mean age = 70.83, SD = 8.17, age range = 56–84 years), 96 typically ageing older European participants (mean age = 66.18, SD = 7.94, age range = 55–83 years), 44 younger European adults (mean age = 21.13, SD = 2.87, age range = 18–26 years), and 94 South Asian older adults (mean age = 67.25, SD = 6.13, age range = 55–79 years). Older and younger European participants were white British or European fluent English speakers with a minimum of 11 years in formal education. The older European participants were recruited from the local community, with the younger adults recruited via Lancaster University’s Research Participant System. The Asian participants were recruited from local Hindu temples located in the northwest of England, who were born in India or East Africa, but had resided in the U.K. for an average of 46.66 years (SD = 5.94). 

The AD and MCI participants were recruited via various NHS sites and memory clinics across the U.K. Participants had received a clinical diagnosis following a full assessment from a dementia specialist. AD participants had a formal diagnosis of dementia due to AD and met the requirements for the American Psychiatric Association’s Diagnostic and Statistical Manual of Mental Disorders (DSM IV) and the National Institute of Neurological and Communicative Disorders and Stroke (NINCDS) for AD. MCI participants met the following criteria [35] and had a diagnosis of dementia due to mild cognitive impairment: (1) subjective reports of memory decline (reported by the individual or caregiver/informant); (2) memory and/or cognitive impairment (scores on standard cognitive tests were >1.5 SDs below age norms); (3) activities of daily living were preserved. The following exclusion criteria were applied: patients with acute physical symptoms, focal cerebral lesions, history of neurological disease (e.g., Parkinson’s disease, multiple sclerosis, epilepsy, amyotrophic lateral sclerosis, muscular dystrophy), cerebrovascular disorders (including ischemic stroke, haemorrhagic stroke, atherosclerosis), psychosis, active or past alcohol or substance misuse/dependence or any physical or mental condition severe enough to interfere with their ability to participate in the study. 

All participants retained the capacity to consent to participation in the study and provided written informed consent. Ethical approval was granted by the Lancaster University Ethics committee and by the NHS Health Research Authority, Greater Manchester West Research Ethics Committee.

### 2.2. Neuropsychological Assessments

The Montreal Cognitive Assessment (MoCA) [36] was administered as an indicator of probable dementia with a score of 26/30 or higher considered normal. The digit [37] and spatial span [37], forward and reverse, were used to estimate short-term memory span and working memory.

### 2.3. Eye Tracking Tasks

#### 2.3.1. Apparatus

Eye movements were recorded using SR EyeLink Desktop 1000 with a sampling rate of 500 Hz. A chin rest was used to minimise head movements, and participants were seated 55 cm away from the computer screen. Prior to the start of each eye tracking task, a 9 point calibration was used. The stimulus was controlled and created via the use of Experiment Builder Software Version 1.10.1630. 

#### 2.3.2. Prosaccade Task

Participants were presented with 36 gap trials followed by 12 overlap trials. A white central fixation point was displayed for 1000 ms, followed by a red target presented randomly at 4° to the left or right for 1200 ms. Participants were instructed to look towards the central fixation point and then when the red target appeared to move their gaze towards it as quickly and accurately as possible. Between trials, a black interval screen was displayed for 3500 ms.

The gap condition included a blank interval screen displayed for 200 ms between the initial appearance of the red target and the extinguishment of the central fixation target. For this condition, the red and white target never appeared on the screen simultaneously (Figure 1). In the overlap condition, the target was presented while the central fixation remained present on the screen for a short period. There was a 200 ms “overlap” in which the target and fixation point were presented simultaneously (Figure 2). After this period, the central fixation was removed, and the target presented singularly for 1200 ms. Previous research [16,18] found that this format works well for patients with neurodegenerative diseases. 

### 2.4. Data Analysis

The raw data were analysed and extracted from the EyeLink using DataViewer Software Version 3.2. The raw data were analysed offline via the use of the software [38]. The software filtered noise and spikes by removing frames with a velocity signal greater than 1500 deg/s or an acceleration signal greater than 100,000 deg^2^/s. The fixations and saccadic events were detected by the EyeLink Parser, and the saccades were extracted alongside multiple spatial and temporal variables. Trials in which the participants did not direct their gaze to the fixation point before the target display were removed. Anticipatory saccades made prior to 80 ms and excessively delayed saccades over 700 ms were also filtered from the data. 

## 3. Results

Linear mixed effects model analyses were carried out using RStudio Version 1.2.5033. The models conducted an analysis of the reaction times in the gap and overlap conditions. The “gap effect” value was calculated by subtracting the individuals mean latency in the gap condition from the overlap condition mean latency. The linear mixed effects model also determined the group effects of disease, ageing, and ethnicity. Two participants in the MCI group were excluded from the subsequent analyses due to their mean reaction times in the prosaccade gap and overlap condition being greater than two standard deviations away from the mean. 

### 3.1. Neuropsychological Tests

A linear mixed effects model was conducted to analyse the performance on the neuropsychological tests. Table 1 shows that there was the expected effect of disease on the MoCA test, with lower scores for the AD participants compared to older European participants, β = 6.90, *t* (257) = 7.25, *p* < 0.0001. AD participants scored significantly lower than the MCI participants, β = 2.83, *t* (257) = 2.72, *p* = 0.007 (see Table 2). The European older adults produced higher MoCA scores than the MCI group (β = −4.06, *t* (257) = −5.13, *p* < 0.001) and unexpectedly the South Asian group (β = −6.89, *t* (257) = −7.25, *p* < 0.001). This difference could be due to the combination of culturally inappropriate test items, linguistics and other cultural factors. There were no significant differences in the MoCA between the healthy European older and younger adults. 

The digit span test (total score forward and backward) revealed that AD participants had a significantly lower mean score than the older European participants, β = 2.38, *t* (254) = 2.46, *p* = 0.015. No significant differences were found between the AD and MCI group (see Table 2). The older South Asian participants had significantly lower digit span than the older European participants (β = −4.45, *t* (254) = −6.49, *p* < 0.001). A significant difference was found between younger and older European adults, with a higher mean digit span score for the younger adults (β = 2.25, *t* (254) = 2.45, *p* = 0.015).

Table 1 shows the results from the spatial span (forward and backward) and revealed that, as expected, the AD participants scored significantly lower compared to the older European participants, β = 2.41, *t* (242) = 3.99, *p* < 0.001. The AD participants had a significantly lower spatial span score than MCI participants, β = 1.50, *t* (242) = 2.32, *p* = 0.021. The findings revealed an ageing effect with young adults producing significantly higher spatial span scores than the European older adults (β = 3.53, *t* (242) = 6.15, *p* < 0.001). The older South Asian participants scored significantly lower than older European participants, β = −1.39, *t* (242) = −3.17, *p* = 0.002 (Table 2). 

### 3.2. The “Gap” Effect

Figure 2 shows the relative shift in the latency distributions in the gap and overlap trials for each of the participant groups. A linear mixed model analysis was conducted to analyse the reaction times in relation to the participant groups. The overlap condition yielded significantly longer reaction times overall, compared to the gap condition, β = 108.21, *t* (8881) = 57.33, *p* < 0.0001 (Figure 2). The “gap effect” was therefore evident in all groups, with significantly faster reaction times in the gap condition, compared to the overlap condition. 

### 3.3. Attentional Disengagement: Effects of Ageing 

The older European participants’ and the younger European participants’ reaction times were compared in the gap and overlap conditions to determine the effects of age. Table 3 reveals that the mean “gap effect” was significantly smaller in the younger European participants (87 ms) compared to the older European participants (110 ms) β = −23.46, *t* (315) = −2.31, *p* = 0.022. Results showed baseline differences in prosaccades with younger European participants having significantly faster reaction times in the gap (β = −8.22, *t* (4624) = −2.70, *p* = 0.007) and overlap conditions (β = −38.46, *t* (4257) = −6.81, *p* < 0.001) compared to older European participants. This indicated that older European participants showed a greater difficulty in disengaging attention from the central fixation in comparison to the younger adults in addition to a general slowing in prosaccades. 

### 3.4. Attentional Disengagement: Effects of Cognitive Impairment 

Table 4 reveals that there was a significant difference between the AD and older European participants’ saccadic reaction times in the gap condition (β = −10.20, *t* (4624) = −2.92, *p* = 0.004). There was no significant difference in reaction times in the overlap (β = −2.41, *t* (4257) = −0.361, *p* = 0.718) condition. There was also no significant difference between the “gap effect” between the conditions (β = −4.29, *t* (315) = −0.376, *p* = 0.707). Similarly, there were no significant differences in reaction times in these conditions when comparing the AD group with the MCI group (Table 4). There were no significant differences between the MCI and European older controls in the overlap condition; however, in the gap condition, MCI participants revealed a significant increase in mean saccadic reaction times compared to the European older controls. Thus, prosaccades and the “gap effect” were generally well preserved in people with AD and MCI.

### 3.5. Attentional Disengagement: Ethnicity/Cultural Effects

The older European group was contrasted with the South Asian older adults to determine the effects of ethnicity on prosaccade reaction times and the “gap effect”. The results shown in Table 4 revealed that the European older group generated faster reaction times compared to the South Asian older group (β = 15.78, *t* (4624) = 6.28, *p* < 0.001) in the gap condition and overlap conditions (β = 9.95, *t* (4257) = 2.42, *p* = 0.016). There was no difference in the proportion of the “gap effect” between the groups (Table 4). 

### 3.6. Correlations

The neuropsychological measures of memory yielded separate scores: forward, backward and total scores for digit and spatial memory, thus six measures of memory in total. The forward recall score yielded an index for memory span, whilst the backward recall score yielded a more direct measure of working memory, since it relied not simply on pure recall, but also cognitive manipulation of the items in short-term memory. Table 5 reveals that there was a significant negative correlation for backward spatial memory and the gap-effect for the MCI group, such that people with longer attentional disengagement reactions times were associated with lower spatial working memory. Interestingly, this relationship was not evident for digit span, which probed verbal working memory. Curiously, this relationship appeared to be specific to the MCI group, although it was not clear why this relationship was specific to MCI. For many participants, MCI was an intermediate transition state between healthy cognition and Alzheimer’s disease. A significant proportion, but by no means all, will unfortunately go on to develop full-blown AD, although we do not yet have a reliable predictive behavioural measure of those people with MCI who will progress to AD. It appears that during this transition period, attentional disengagement may provide a useful index of the decline in working memory, and the progression from MCI to AD. Longitudinal studies will be required to determine the validity of this hypothesis.

## 4. Discussion

This study revealed that the “gap effect” was well preserved in AD and MCI participants. Participants produced significantly faster reaction times when performing pro-saccadic eye movements during the gap condition compared to the overlap condition. Moreover, the effect was robust across both ethnic/cultural groups explored in this study. 

### 4.1. What Does the Gap Effect Reveal about the Integrity of the Alzheimer Brain?

The neurophysiological networks that regulate the control of saccadic eye movements are relatively well understood. The saccadic eye movements are generated by precise reciprocal activation of saccade-related neurons and the inhibition of fixation neurons in the superior colliculus [39,40]. According to the Findlay and Walker [41] model, the removal of the fixation target leads to a reduction in the activation of the fixation units, which releases the saccade from inhibition, and this is reflected by the reduction in reaction times. When the fixation point remains on, the fixation units are tonically active, and the move units are inhibited, causing a delay in the initiation of a saccade. This network is clearly well preserved in early and late stages of the disorder. In previous work, we examined inhibitory control saccades extensively using the antisaccade task. In contrast to the gap and overlap task, the anti-saccade task requires that the observer looks away from the object, in the opposite direction, and is one of the most widely used paradigms assessing inhibitory control in both healthy individuals and clinical disorders [42,43]. These studies have shown that people with dementia generate a high proportion of uncorrected prosaccade errors towards the target in the antisaccade task that correlates with the severity of the dementia [16]. In contrast, when healthy participants make errors, they are normally rapidly corrected, although both AD and MCI adhere to the principle that the frequency of past errors predicts the probability of future errors [44]. People with amnesic MCI are at a greater risk of progressing to dementia [45,46,47]. Recently, our lab has shown that these errors are also evident in amnesic MCI to a greater extent than non-amnesic MCI participants [20]. We have argued that this error correction implicates a neural network that includes the anterior cingulate. Together with this work, the current evidence of the preservation of the attentional disengagement [16] will help to increase our understanding of the specificity of oculomotor impairment in AD and undermine the idea that the source of the uncorrected errors can be attributed to the inability to disengage attention from the prepotent target. Rather, the inhibition appears to be directly linked to top-down inhibitory control and working memory [16,17]. Clearly, there is a dissociation of impairment of the oculomotor pathways in AD. Evidence from this study revealed a preservation of the superior colliculus pathway, while converging evidence from previous and more recent work [20] indicated that other centres of the network, including the anterior cingulate, that mediate top-down inhibitory control and error monitoring are affected early in the course of the disease [21]. 

### 4.2. Ageing 

Another key finding was a strong ageing effect on the saccadic reaction times. Although all participant groups displayed the gap effect, the younger adults revealed a significantly faster mean reaction times in the overlap and gap conditions than the older adults. Previous research has reported that eye movements are susceptible to ageing effects, in particular reductions in processing speed, spatial memory and inhibitory control [48,49,50,51]. Crawford et al. [17] reported that the “gap effect” increased in older adults compared to younger adults, suggesting that the changes in the attentional engagement are associated with normal ageing, rather than AD. The older adults are apparently more dependent on the removal of the central stimulus to facilitate the shift of attention from fixation and therefore showed a larger benefit following the removal of the fixation point in the gap condition compared to the younger adults. One possible explanation is that may be due to an age-related decrease in the reciprocal inhibitory activity of the fixation and move units [41]. 

### 4.3. Ethnicity 

As outlined above, the European and the South Asian older adults both demonstrated the gap effect, with significantly faster prosaccades in the gap condition compared to the overlap condition. Clear differences between the groups emerged in the saccade reaction times, specifically for the saccade gap and overlap conditions, with South Asian adults presenting slower saccade reaction times. This raises the possibility that the south Asian group may have a lower proportion of fast and express saccades in the gap and overlap tasks. Express saccades [52,53] are fast reaction time saccades (80 ms−130 ms) with frequencies that vary across cultural groups. The frequency of express saccades was reduced in the overlap task, because the temporal overlap of the fixation-point and the target often inhibited the prosaccade, which may have reduced the difference in the overlap condition. Knox, Amatya, Jiang and Gong [33] demonstrated that Chinese participants showed a higher proportion of express saccades compared to U.K. participants. Clearly, saccade performance can differ across different cultural groups. If express saccades were a contributory factor to the faster saccade latencies of the European group in the gap condition, this would explain the convergence of saccade latencies for the groups in the overlap condition. However, the combined group latency distributions in Figure 2 suggests that this hypothesis may be flawed and cannot account for the group differences in the gap task and overlap task, although this would be best examined within a design with a larger number of trials, with the distributions of individual participants. 

Previous research has also shown differences in eye movements across different cultural and ethnic groups [33,54,55]. Chua, Boland and Nisbett [56] found differences in scan patterns between native Chinese and native English-speaking participants when assessing visual scenes. English participants tended to look first at the foreground object and had an increased number of fixations then Chinese participants, predominantly focused on the background visual areas of the scene. Eye movements are clearly not homogeneous; culture and ethnicity factors can influence specific features of eye movement control. 

Knox and Wolohan [57] examined saccades in European, Chinese and U.K.-born Chinese participants who shared similar cultural experiences as the European group. The study investigated whether the differences in the saccadic eye movements of the Chinese and European groups resulted from cultural or culture-unrelated factors. The Chinese participants showed similar pattern results irrespective of the culture exposure. Therefore, cultural differences cannot be the primary cause of the difference in oculomotor characteristics. Although the principal explanatory factors of these differences in oculomotor systems is unclear, they are possibly related to a combination of genetic, epigenetic and environmental factors [58]. A recent study showing very clear differences in the post-saccadic oscillations of Chinese-born and U.K.-born undergraduates concluded that “..genetic, racial, biological, and/or cultural differences can affect the morphology of the eye movement data recorded and should be considered when studying eye movements and oculomotor fixation and saccadic behaviors” [34]. Although there has been increasing eye-tracking research with Chinese participants, research with South Asian populations has been sparse. We hope that this work will encourage future studies to help redress this void.

## 5. Conclusions

Research scientists have tended to focus on the memory, intelligence and other mental skills that degenerate in AD and understandably have paid less attention to those equally important cognitive functions that may be well preserved. A better understanding of preserved functions in the disease will help to develop potential new early intervention strategies in the treatment of the disease that may improve mental functions and delay the progression of the disease. Patients with AD show large individual differences in the profile of scores across both traditional cognitive assessment and measures of saccadic eye movement. Therefore, in our recent work [18,21], we developed a profile measure of z-scores for each test that captures a patient’s performance across a range of measures in relation to the normative scores. This approach takes full advantage of the extensive range of saccadic eye movement parameters to assess and monitor cognitive changes in the evolution of AD and will enhance specificity and sensitivity as a diagnostic tool. 

Further, this study demonstrated that prosaccades can be susceptible to disease, ageing and ethnicity effects, and therefore, future research should strive to include non-WEIRD participant groups to create a more comprehensive understanding of the effect and its robustness and generalizability.

## Figures and Tables

**Figure 1 brainsci-10-00461-f001:**
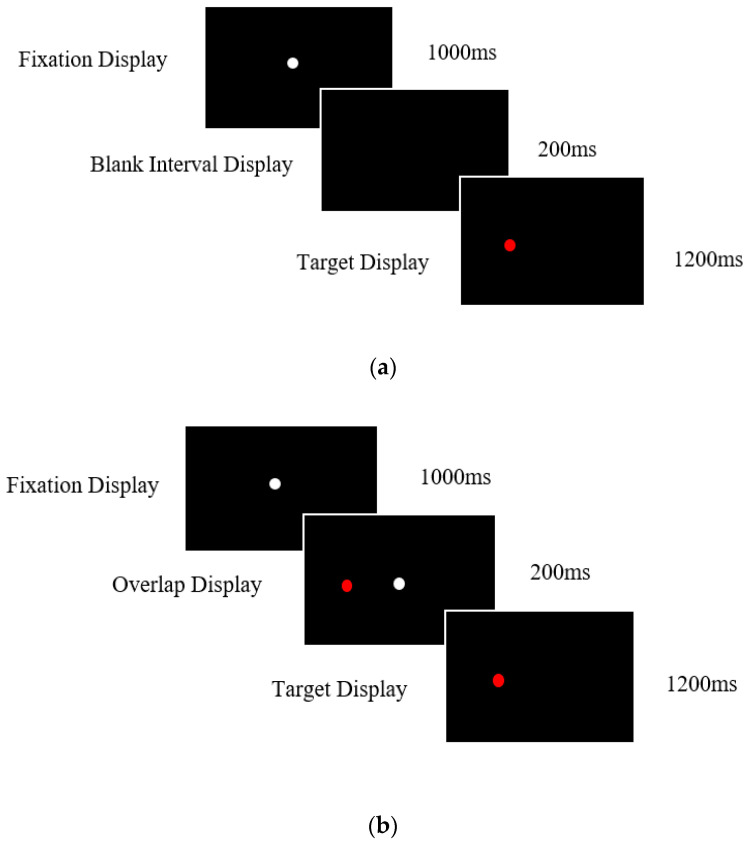
**Gap and Overlap Displays** (**a**) Timings and sequence of the prosaccade task gap condition. The gap condition facilitates the disengagement of visual attention prior to the target’s presentation due to the removal of the central fixation point. (**b**) Timings and sequence of the pro-saccade task overlap condition. The central fixation point remains on for a short period when the target is displayed. This results in a delay in the disengagement of attention, resulting in longer mean saccade reaction times.

**Figure 2 brainsci-10-00461-f002:**
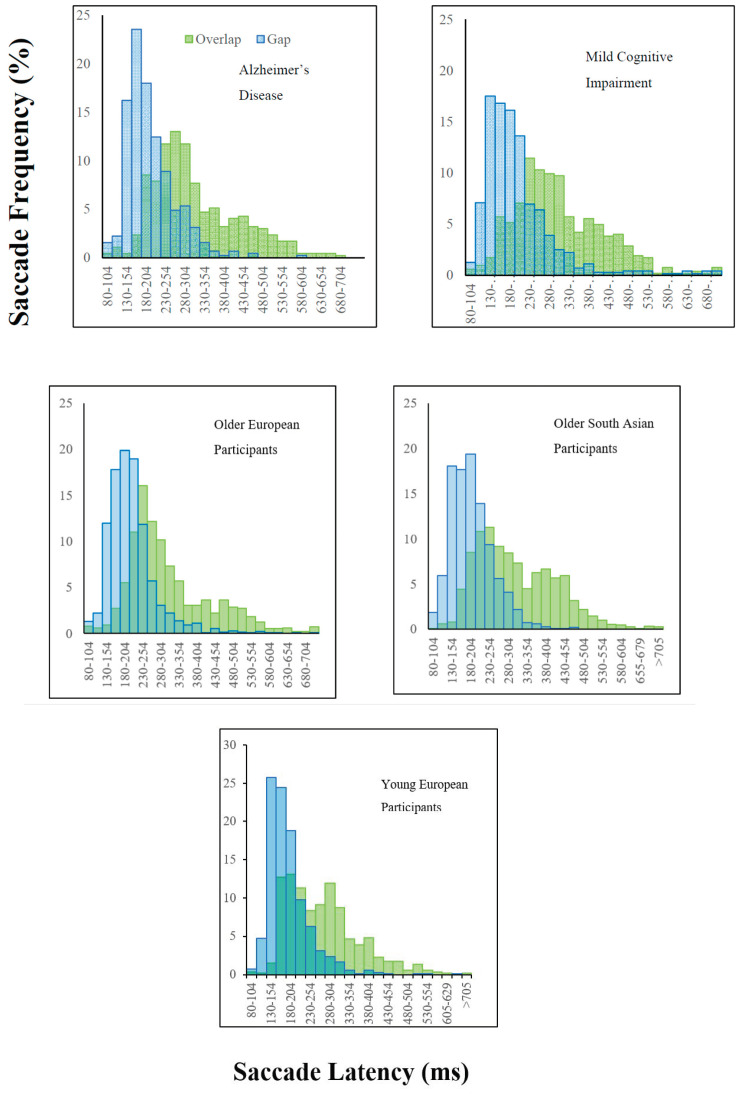
Histograms displaying a shift in the distribution of saccade latencies in the gap condition (blue) compared to the overlap condition (green) for the participant groups: Alzheimer’s disease, mild cognitive impairment, older and younger European participants and older South Asian participants.

**Table 1 brainsci-10-00461-t001:** Table displaying mean reaction times and standard deviations for the neurological assessments. MCI—mild cognitive impairment.

	Older European Participants		Older South Asian Participants		Alzheimer’s Disease		MCI		Young European Participants	
	M	SD	M	SD	M	SD	M	SD	M	SD
**MoCA**	27.80	2.04	22.04	4.99	20.19	5.45	22.98	5.40	28.14	1.94
**Digit Span Task**	17.91	4.60	13.27	3.71	15.23	4.56	15.95	4.12	19.86	4.33
**Spatial Span Task**	13.89	2.44	12.47	2.24	11.42	3.75	12.93	3.08	17.38	2.08

Note: dependent variable: total task score.

**Table 2 brainsci-10-00461-t002:** Table displaying post hoc comparisons for the neurological assessments.

	Post Hoc Contracts (*p* Values)				
	Disease Effects			Ageing Effects	Ethnicity Effects
	AD vs. OEP	AD vs. MCI	MCI vs. OEP	OEP vs. YEP	OEP vs. OSP
**MoCA**	<0.001 *	0.007 *	<0.001 *	0.025	<0.001 *
**Digit Span Task**	0.015 *	0.496	0.043	0.015 *	<0.001 *
**Spatial Span Task**	<0.001 *	0.021 *	0.077	<0.001 *	0.002 *

AD—Alzheimer’s disease; MCI—mild cognitive impairment; OEP—older European participants; OSP—older South Asian participants. YEP—young European participants. * Significant at *p* < 0.05.

**Table 3 brainsci-10-00461-t003:** Table displaying mean reaction times and standard deviations for the prosaccade task gap and overlap conditions.

	Older European Participants *N* = 96		Older South Asian Participants *N* = 94		Alzheimer’s Disease *N* = 32		MCI *N* = 45		Young European Participants *N* = 44	
	M	SD	M	SD	M	SD	M	SD	M	SD
**Gap**	195	38.87	212	37.06	206	30.93	200	42.18	185	31.60
**Overlap**	305	75.06	315	75.06	312	51.32	310	66.86	272	58.83
**Gap Effect (ms) (Overlap-Gap)**	110	57.30	103	58.66	106	48.06	110	59.54	87	48.53

**Table 4 brainsci-10-00461-t004:** Table displaying post hoc comparisons for the prosaccade task gap and overlap conditions.

	Post Hoc Contracts (*p* Values)				
	Disease Effects			Ageing Effects	Ethnicity Effects
	AD vs. OEP	AD vs. MCI	MCI vs. OEP	OEP vs. YEP	OEP vs. OSP
**Gap**	0.004 *	0.161	<0.001 *	0.007 *	<0.001 *
**Overlap**	0.718	0.972	0.706	<0.001 *	0.016 *
**Gap Effect** **(Overlap-Gap)**	0.707	0.885	0.803	0.022 *	0.383

AD—Alzheimer’s disease; MCI—mild cognitive impairment; OEP—older European participants; OSP—older South Asian participants. YEP—young European participants. * Significant at *p* < 0.05.

**Table 5 brainsci-10-00461-t005:** Correlations of prosaccade conditions and neuropsychological tests.

Variable		MoCA	Digit Span Total	Digit Span Forward	Digit Span Backward	Spatial Span Total	Spatial Span Forward	Spatial Span Backward
**Gap Effect** **(Overlap-Gap)**	AD	−0.063	0.120	0.048	0.169	0.157	0.242	0.058
	MCI	−0.096	−0.076	−0.118	−0.015	−0.168	0.021	−0.318 *
	OEP	−0.095	−0.004	−0.014	−0.031	−0.014	0.025	−0.045
	OSP	0.213	0.153	0.144	0.126	−0.024	−0.028	−0.013
	YEP	0.147	−0.013	0.070	−0.088	0.074	0.071	0.043

AD—Alzheimer’s disease; MCI—mild cognitive impairment; OEP—older European participants; OSP—older South Asian participants. YEP—young European participants. * Significant at *p* < 0.05.

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
