# Peer review of "The Disengagement of Visual Attention: An Eye-Tracking Study of Cognitive Impairment, Ethnicity and Age"

_brainsci, 2020, doi:10.3390/brainsci10070461_

Round 1
Reviewer 1 Report
This paper presents a valuable dataset on performance of gap / overlap saccades tasks in several different populations of younger and older adults. Overall the study is sound and it is important to have the paper in the public domain, although currently the it reads as a bit of a data dump, with no really interesting ideas explored in the discussion sections. I think more could be made of the results and a better effort made in the writing to present a story. I make some suggestions below organised by section:
Introduction
The importance of the research is set out in the opening paragraph in which it is highlighted that simple saccade tasks may have potential in the assessment and diagnosis of neurodegenerative diseases. The 2nd paragraph deals with attentional engage/disengage processes but it quite long and I suggest could be scaled back. If there is any evidence from non-oculomotor research that attentional engage / disengage functions are affected or not in alzheimers and MCI then this should also be included in the introduction.
Gap/Overlap Prosaccade tasks can then be introduced as a potentially useful way to assess attentional functions in patients as an alternative to neuropsychological tests. If they are to be useful then there should be differences between MCIs/Alz and Older controls in the eye movement tasks and correlations between eye movement tasks and other cognitive tests.
The variation in eye movements by ethnicity then comes in as a secondary question of interest and of particular importance to establishing older adult normative performance.
Methods
The participants completed 36 gap trials but only 12 overlap trials. Why was there a difference in trial numbers? Couldn't this have potentially affected performance? How do the results differ if you just compared the first 12 gap trials with the 12 overlap trials?
Results
I can't see any correlations between the cognitive tests and the eye movement reaction times. I suggest you investigate correlations between the spatial / digit span, MCoA and saccade tasks for the different groups.
Two of the MCI participants were excluded for having very long reaction times. But this might be what you'd expect if the the eye movement tests might be sensitive to onset of neurodegenerative changes that were occuring in some of the MCI participants. So I think excluding these participants is "throwing the baby out with the bathwater". Are the long RTs in these participants genuine or artefacts of bad eye tracker calibration? Where do they lie on the normative curve for older adults? Did they have particularly low cognitive test scores?
Discussion
I expect there will be more results of interest to discuss when you have done the correlations, but the conclusions with respect to the hypotheses are:
- Gap / Overlap tasks are not particularly promising for discriminating between healthy older adults and dementia. Although the discussion could explore what eye movement tests are in more detail (e.g. anti saccades) and what the implications are in terms of what apsects of attentional / cognitive / oculomotor control processes are affected in early stage Alzheimers and MCI.
- There are differences in eye movement reaction times between "caucasians" and "south asians". Currently the discussion of this effect (Section 4.2 paragraph 1) is confused. It states that "the hypothesis"(sic) about express saccades is "flawed" but doesn't provide an alternative. It looks to me that the South Asian group has a longer tail on their reaction time distribution for the overlap condition, but why would this be? Could other health issues prevalent in this group (e.g. heart disease; diabetes?) explain the differences in attentional functions? Direct effects of genetic and cultural factors seem quite unlikely to be primary influences before other health effects have been excluded. The group also seems to score lower on the cognitive tests. Might oculomotor control differences explain this? Spatial working memory tasks in particular require coordination of complex oculomotor control strategies (Hodgson et al, 2019; 2020).
Finally I'm not sure about the term "Caucasian" for White Europeans. I it came out of US TV cop shows rather than being a genuine scientific term. How many of your participants were actually from Caucasia? Suggest a geographical term as you have used for "South Asians" i.e. Northern European or just European.
Reviewer 2 Report
Dear Authors,
I read with great interest Your manuscript submitted for publication in Brain Sciences.
The disengagement of visual attention as early warning sign was reported in the literature, and Your study added a further useful contribution.
These are my comments and suggestions :
> it was unclear whether the number of participants was reached based on who met inclusion and exclusion criteria, or in what other way. Who responded firstly.....for example ?
> The effect of ageing on attentional disengagement should be better explained and discussed.
> Some exclusion criteria had to be more detailed. For instance, what did You mean for "cardiovascular " or "cerebrovascular" disease ?
> What are the consequences of Your data in clinical practice ? In Discussion, it is not clear.
